# Complete Chloroplast Genome Sequence of Endangered Species in the Genus *Opisthopappus* C. Shih: Characterization, Species Identification, and Phylogenetic Relationships

**DOI:** 10.3390/genes13122410

**Published:** 2022-12-19

**Authors:** Xinke Zhang, Guoshuai Zhang, Yuan Jiang, Linfang Huang

**Affiliations:** Key Laboratory of Chinese Medicine Resources Conservation, State Administration of Traditional Chinese Medicine of the People’s Republic of China, Institute of Medicinal Plant Development, Chinese Academy of Medical Sciences. Peking Union Medical College, Beijing 100193, China

**Keywords:** chloroplast genome, *Opisthopappus* C. Shih, comparative analysis, phylogenetic analysis, molecular marker

## Abstract

*Opisthopappus* C. Shih is a rare genus of the Asteraceae family native to the Taihang Mountains in China. Due to the narrow distribution area, poor reproduction ability and human harvesting, *Opisthopappus* is threatened by extinction. However, the limited genetic information within *Opisthopappus* impede understanding of the conservation efforts and bioprospecting. Therefore, in this study, we reported the complete chloroplast (cp) genome sequences of two *Opisthopappus* species, including *Opisthopappus taihangensis* and *Opisthopappus longilobus*. The cp genomes of *O. taihangensis* and *O. longilobus* were 151,117 and 151,123 bp, which contained 88 protein-coding genes, 37 tRNA genes, and 8 rRNA genes. The repeat sequences, codon usage, RNA-editing sites, and comparative analyses revealed a high degree of conservation between the two species. The *ycf*1 gene was identified as a potential molecular marker. The phylogenetic tree demonstrated that *O. longilobus* was a separate species and not a synonym or variety of *O. taihangensis.* The molecular clock showed that two species diverge over a large time span, *O. longilobus* diverged at 15.24 Mya (Million years ago), whereas *O. taihangensis* diverged at 5.40 Mya We found that *Opisthopappus* and *Ajania* are closely related, which provides new ideas for the development of *Opisthopappus.* These results provide biological information and an essential basis to understand the evolutionary history of the *Opisthopappus* species, which will aid in the future the bioprospecting and conservation of endangered species.

## 1. Introduction

*Opisthopappus* C. Shih belonging to the family Asteraceae was generally considered as including two species *O. taihangensis* and *O. longilobus* [1], and it is native to the Taihang Mountains in China, growing on the cliffs at an altitude of 1000 m [2,3]. The species of this genus are rich in phenolic compounds and flavonoids and have been consumed as medicine and food in the origin areas [4]. Based on the narrow distribution area, poor reproduction ability and human harvesting, *Opisthopappus* is now in an endangered state and has been listed as a rare and endangered species [5]. Furthermore, the species classification within *Opisthopappus* is not clear. According to the morphological and molecular evidence, *O. taihangensis* and *O. longilobus* were previously classified as the same species [6]. The phylogenetic relationships between these two species were unclear due to limited resolving power of morphological methods and detection capacity of unique nuclear gene sequences [7].

In green plants, cp plays an important role in photosynthesis [8]. The cp genomes show high conservatism regarding genome size, structure, gene content, and organization [9,10]. Therefore, the cp genome has been used as an ideal technology for the investigation of phylogenetic analyses, plant molecular identification, and genetic diversity evaluation. Recently, the whole cp genome sequences can be used as a plant super-barcode for discriminating closely related species in some taxa [11,12]. However, there are few available on the cp genome of *Opisthopappus*, only the cp of *O. taihangensis* has been sequenced [13], and the cp genome information is still missing in *O. longilobus*. In addition, the comparative genomics within the genus *Opisthopappus* have not been studied, which limits the phylogenetic and genetic diversity studies in this genus.

In this study, the complete cp genomes of two *Opisthopappus* species were sequenced and annotated for the first time. We analyzed the general characteristics and compared the structural characteristics to determine the origin and phylogenetic relationships. These results provide abundant genetic information of the genus *Opisthopappus* and formulate effective conservation and molecular identification approaches for the crucial and endangered medicinal plants.

In this study, the complete cp genomes of two *Opisthopappus* species were sequenced and annotated. We analyzed the general characteristics and compared the structural characteristics to determine the origin and phylogenetic relationships. The results of this study will provide abundant genetic information on the genus *Opisthopappus* and formulate effective conservation and molecular identification approaches for the crucial and endangered medicinal plants.

## 2. Materials and Methods

### 2.1. Plant Materials, DNA Extraction, and Sequencing

The fresh leaves of *O. taihangensis* and *O. longilobus* were collected from Xinxiang City (Henan, China, 35°43′ N, 113°36′ E) on 20 August 2020, which were identified by Professor Linfang Huang. Specimen of the two *Opisthopappus* species were stored at the Herbarium of the Chinese Academy of Medical Science & Peking Union Medicinal College, accession numbers: 20200820A1 and 20200820A2. Total DNA was extracted using the CTAB (cetyltrimethyl ammonium bromide) method [14]. The DNA sequencing was performed using the Illumina NovaSeq system (Illumina, San Diego, CA, USA).

### 2.2. Cp genome Assembly and Annotation

The cp genomes were assembled from clean reads using NOVOPlasty v.3.8.3 [15], and then, the assembled file was annotated by CpGAVAS2 web service (accessed on 20 May 2022) [16]. The tRNA genes were identified using tRNAscan-SE program v.1.3.1 [17]. The annotated whole genome sequences were submitted to GenBank (accession numbers: MZ779049, MZ779050). Finally, circular maps of cp genomes were drawn by the OGDRAW program [18]. 

### 2.3. Repeat Element Analysis

The repeat sequences were performed using CPGAVAS2 web service. Tandem repeats were identified by Tandem Repeats Finder v.4.09 [19], with specific screening parameters of 2 for matches and 7 for mismatches and indels. The similarity cutoff value among the repeat units was set at 70%. The simple sequence repeat (SSR) was predicted using MISA (https://webblast.ipkgatersleben.de/misa/) (accessed on 25 May 2022) [20]. The scattered repetitive sequences were identified by Vmatch (http://www.vmatch.de) (accessed on 31 May 2022). The location and size of repeat sequences were recognized using REPuter (https://bibiserv.cebitec.uni-bielefeld.de/reputer) (accessed on 25 May 2022) [21].

### 2.4. Codon Usage and RNA-Editing Sites Analysis

Phylosuite v.1.2.2 was used to extract the protein-coding genes, and then, CodonW v.1.4.2 was used to calculate codon usage and relative synonymous codon usage (RSCU) values [22,23]. Lastly, the RSCU values were displayed in the form of a heatmap by TBtools software v.1.6 [24].

In addition, the predictive RNA Editor for Plants (PREP) suite was used at the cutoff value of 0.8 to analyze the potential RNA editing sites in the protein-coding genes of the cp genomes [25,26].

### 2.5. Phylogenetic and Divergence Time Analysis

In total, 20 complete cp genomes of subtribe Chrysantheminae were downloaded from the NCBI for phylogenetic analysis. In addition, *Xanthium spinosum* (NC_054222) was used as the outgroup. These cp genome sequences were aligned by MAFFT v. 7.307 [27]. Subsequently, the alignment was conducted with the maximum likelihood (ML) method using RAxML v. 8.2.4 [28]. The detailed parameters were set to “raxmlHPC-PTHREADS-SSE3 -fa -N 1000 -m GTRGAMMA -x 551314260 -p 551314260 -o Xanthium_spinosum_NC_054222 -T 20”. The reliability of the phylogenetic tree was assessed using bootstrap method with 1000 replications.

The molecular clock tree was constructed using MEGA based on an ML method to estimate the origin and divergence times of *Opisthopappus* and related genus [29]. The relevant divergence times can be found in the TimeTree Resource database (http://www.timetree.org/) (accessed on 25 May 2022) [30]. Furthermore, *X. spinosum* was selected as outgroup of the phylogeny. The temporal constraints were derived from the TimeTree Resource. NODE TIME found the divergence time between two genera.

### 2.6. Genome Comparison

Sliding window analysis was conducted to assess the nucleotide diversity (Pi) values of the cp genomes by DnaSP v6 (window length = 1000 bp, step size = 300 bp). The four complete cp genome sequences (*O. taihangensis* (NC_042787), *O. taihangensis*, *O. longilobus*, and *Ajania pacifica* (Nakai) K. Bremer et Humphries (NC_050690)) were compared by mVISTA (http://genome.lbl.gov/vista/mvista/) (accessed on 28 May 2022) [31]. The cp genome sequence of *O. taihangensis* (NC_042787) was downloaded from NCBI and used as a reference sequence for annotation [13]. Furthermore, the cp genetic architecture in LSC/IRs and SSC/IRs borders of four species were identified and compared using IRscope (http://genocat.tools/tools/irscope.html) (accessed on 28 May 2022) [32]. 

## 3. Results and Discussion

### 3.1. Characteristics of O. taihangensis and O. longilobus cp Genomes

The total length of the cp genomes of *O. taihangensis* and *O. longilobus* were 151,117 and 151,123 bp (Figure 1). The two species cp genomes showed the commonplace quadripartite construction comprising an LSC area (82,901 and 82,895 bp), an SSC locale (18,306 and 18,320 bp), and a pair of IRs districts (24,955 and 24,954 bp). In addition, the overall GC contents were 37.46% and 37.44%, which were unevenly distributed across the complete cp genome (Table 1). The IRs were the highest (43.07% and 43.08%), followed by the LSC (35.54% and 35.52%), whereas the SSC region showed the lowest GC content (30.82% and 30.8%). The overall G + C contents in both species were lower than A + T contents, which is a general feature exhibited in many angiosperm species cp genomes sequences.

The cp genomes of *O. taihangensis* and *O. longilobus* were highly conservative, with nearly identical gene content, gene order, and no structural reconfigurations. In total, 133 genes were predicted in two species, consisting of 88 protein-coding genes, 37 tRNA genes, and 8 rRNA genes. Most of the genes of two cp genomes were generally classified into three categories (Table 2), including self-replication, photosynthesis-related, and other function genes, respectively. *Opisthopappus* had lost some genes during evolution, like most angiosperms, such as *chI*B, *chI*L, and *ycf*68. [33,34]. The complete cp genomes of the two species were highly conserved intron number and type. In total, 19 genes contained introns. Among those genes, eight genes (*rpl*2 (×2), *ndh*B (×2), *trn*A-UGC (×2), and *trn*E-UUC (×2)) were located in the IR region, and the remaining ten genes (*rps*16, *atp*F, *rpo*C1, *ycf*3, *clp*P, *pet*B, *trn*K-UUU, *trn*S-CGA, *trn*L-UAA, and *pet*D) were located in the LSC region. However, *ndh*A was the only gene found in the SSC region. Among them, the *clp*P and *ycf*3 had two introns (Appendix A).

To investigate the genomic differences, we compared the cp genome features of NC_042787 and MZ779049. As shown in Table 3, the gene content and organization of MZ779049 were similar to NC_042787. However, the cp genome of MZ779049 had the longer genome length and higher number of genes, which indicated that MZ779049 showed more complete cp genome.

### 3.2. Repetitive Sequence

SSRs consist of tandem short repeat units and distribute widely throughout the cp genome. In this study, the total number of SSRs detected in the cp genome sequences of *O. taihangensis* and *O. longilobus* were 42 and 44 (Figure 2a). Among them, mononucleotide SSRs were the most abundant, followed by both dinucleotides and tetranucleotides. Other types of SSRs were not detected. Moreover, intergenic regions had more abundant SSRs than the protein-coding regions, which is similar to those in most angiosperms’ cp genome [35]. 

In addition, the long repeat sequences of the cp genomes in the two species were similar. *O. taihangensis* showed 20 forward and 22 palindromic repeats, while *O. longilobus* displayed 20 forward and 21 palindromic repeats (Figure 2b). These repetitive sequences provide meaningful clues for the studies of genetic diversity and the development of molecular markers.

### 3.3. Codon Usage Analysis and RNA-Editing Sites

The length of the protein-coding genes regions in *O. taihangensis* and *O. longilobus* were 78,636 and 78,624 bp, which were used to calculate RSCU values. An RSCU value greater than 1 indicates high frequency usage. As shown in Figure 3, these encoded protein sequences consisted of 21 amino acids. High RSCU values are represented by red, and low RSCU values are indicated by blue. The heatmap shows that 30 codons were used frequently in the two species. With the exception of UUG, all preference codons finish in purines (A/U) because of the nature of the A/T-rich cp genome.

Additionally, in the cp genomes of *O. taihangensis* and *O. longilobus,* latent RNA editing sites were discovered for 18 genes. In total, 47 RNA editing sites were identified in the two species. Among the two species cp genomes, the gene with the most RNA editing sites was *ndh*B. The majority of the detected RNA-editing sites were at the second codon position and included cytosine to uracil (C-U) conversions. Serine to leucine (S-L) conversions were the most frequent amino acid conversions. In addition, a large number of RNA-editing sites resulted in modifications to amino acids in hydrophobic products such as phenylalanine (F), tyrosine (Y), leucine (L), and valine (V) (Appendix A).

### 3.4. Phylogenetic Analysis and Divergence Time Analysis

The ML phylogenetic tree (Figure 4) reveals that *Opisthopappus* was more closely related to *Chrysanthemum* and *Ajania* in the subtribe Chrysantheminae. They formed a strongly supported sister relationship with *Artemisia*. Furthermore, *O. taihangensis* sequenced in this study was clustered with the published *O. taihangensis* (NC_042787). *O. longilobus* was more closely related to *A. pacifica* than to *O. taihangensis.* This indicated that *O. longilobus* was a separate species and was not a synonym or variety of *O. taihangensis*. Closely related plants are chemically similar and may have the same pharmacological properties. Moreover, plants are phylogenetically related to each other. Therefore, ethnobotanists have used a range of phylogenetic methods for bioprospecting [36]. From a previous study, *Ajania* has anti-inflammatory, anthelmintic and malaria treatment properties [37]. These results provide new ideas for the exploitation of *Opisthopappus*. The cp genomes seemed to provide more solid support for the reconstruction of phylogenetic relationships among these sections. 

A time-divergent phylogenetic tree (Figure 5) was constructed based on an ML tree. Chrysantheminae was inferred to diverge at 28.27 Mya and divided into various genera at 22.9–23 Mya. Furthermore, *O. longilobus* and *Ajania* diverged at 15.24 Mya. However, *O. taihangensis* diverged at 5.40 Mya. The differentiation time span differs by nearly 10 Mya, and the results also support *O. longilobus* as a separate species.

### 3.5. Comparative cp Genomic Analysis

In order to explore the sequence divergence between the two species of *Opisthopappus*, nucleotide diversity (Pi) was estimated to indicate the variability of potential plastid regions. The values of Pi ranged from 0 to 0.01. Among them, 124,370–126,369 bp resign showed high nucleotide diversity (Pi: 0.0067–0.01). This region was identified as the protein-coding region ycf1 (Figure 6).

The phylogenetic tree revealed a tight relationship between *Opisthopappus* and *A. pacifica.* The cp genome sequences of the three species were compared using *O. taihangensis* (NC_042787) as a reference sequence to examine the differences in the cp genome sequences. As shown in Figure 7, the three cp genomes had the lowest variability in the IR region and was relatively high in the LSC and SSC regions, which may be attributable to the presence of highly conserved rRNA sequences in IR regions. In the cp genomes of the three species, the majority of the protein-coding genes were conserved. However, the rpl16 gene had a large mutation. In addition, *O. longilobus* and *Ajania* had greater variation compared to *O. taihangensis*. The variations were predominantly localized in intergenic regions, such as *pet*N-*psb*M, *psb*E-*pet*L, *psb*A-*mat*k, *trn*T-UGU-*trn*L-UAA, and *trn*R-UCU-*trn*G-UCC, which could be considered as possible molecular genetic markers.

Changes in the length of the cp genome are frequently caused by contraction and extension at the boundaries of IR regions [38]. Figure 8 shows the results of IR regions contraction and expansion of the two cp genomes. The *rps*19 gene was located at the LSC/IRb borders, which was mainly located in the LSC at 60–61 bp of the IRb. The gene on the IRb/SSC borders of *Opisthopappus* was *ycf*1; however, there is no gene change on IRb of *A. pacifica*. The *ycf*1 gene, which was primarily found in the SSC at 558 bp of the IRa, served as the SSC/IRa borders for all species. The *trn*H was found in the LSC/IRa borders. However, the *rps*19 was in IRa and was close to the IRa/LSC boundary in *O. taihangensis* (NC_042787). Overall, *A. pacifica* had the highest variability.

## 4. Conclusions

In this study, the cp genome of *O. longilobus* was firstly reported and we resequenced *O. taihangensis*. A comparative analysis with other genomes was also performed. *Opisthopappus* is an endemic cave plant in China, and its harsh growing environment leads it to drought and other stresses. The study of the cp genome can provide more biological information for the sustainability of *Opisthopappus*. Overall, *Opisthopappus* cp genomes had similar structure and gene composition. However, the sliding window results showed that *O. taihangensis* and *O. longilobus* had great variation in *ycf*1, which could be used as a potential barcode to distinguish the two species. Furthermore, we reconstructed a phylogenetic tree by complete cp genomes. The results indicated that *O. longilobus* was a separate species and not a synonym or variety of *O. taihangensis*. We found that *Opisthopappus* and *Ajania* are closely related. The results provide new ideas for the exploitation of *Opisthopappus*. Overall, these results can provide biological information and essential insights into the evolutionary history of the endangered *Opisthopappus* that will contribute to the bioprospecting and conservation of *Opisthopappus* species.

This is the first study to report the cp genome of *O. longilobus*, and we resequenced *O. taihangensis.* A comparative analysis with other genomes was also performed. *Opisthopappus* is an endemic cave plant in China, and its harsh growing environment leads it to drought and other stresses. The study of the cp genome can provide more biological information for the sustainability of *Opisthopappus.* Overall, *Opisthopappus* cp genomes had similar structure and gene composition. However, the sliding window results revealed that *O. taihangensis* and *O. longilobus* had great variation in ycf1, which could be employed as a potential molecular marker to distinguish the two species. Furthermore, we reconstructed a phylogenetic tree by complete cp genomes. The results indicated that O. longilobus was a separate species and was not a synonym or variety of *O. taihangensis*. It is interesting that we discovered the closely relationship between *Opisthopappus* and *Ajania.* The results provide new ideas for the exploitation of *Opisthopappus.* Overall, these results provide biological information and an essential basis to understand the evolutionary history of the *Opisthopappus* species, which will aid in future bioprospecting and conservation of endangered species.

## Figures and Tables

**Figure 1 genes-13-02410-f001:**
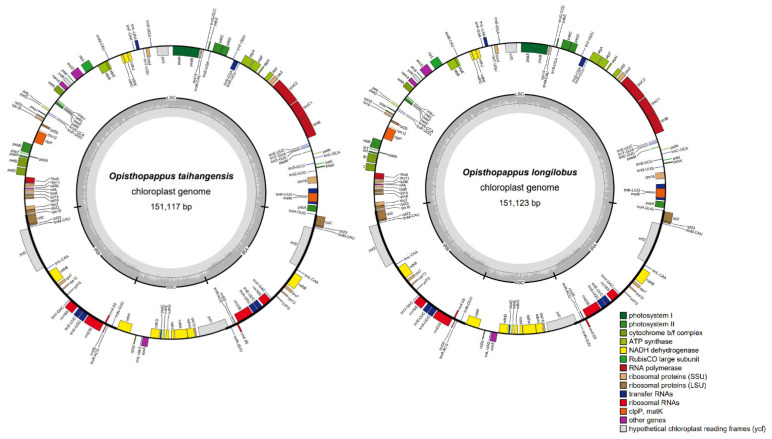
Gene map of *O. taihangensis* and *O. longilobus*. Genes lying outside the outer circle are transcribed in a counter-clockwise direction, and genes inside this circle are transcribed in a clockwise direction. The colored bars indicate known protein-coding genes, transfer RNA genes, and ribosomal RNA genes. The dashed, dark grey area in the inner circle denotes GC content, and the light grey area indicates genome AT content. LSC, large single-copy, SSC, small single-copy; IR, inverted repeat.

**Figure 2 genes-13-02410-f002:**
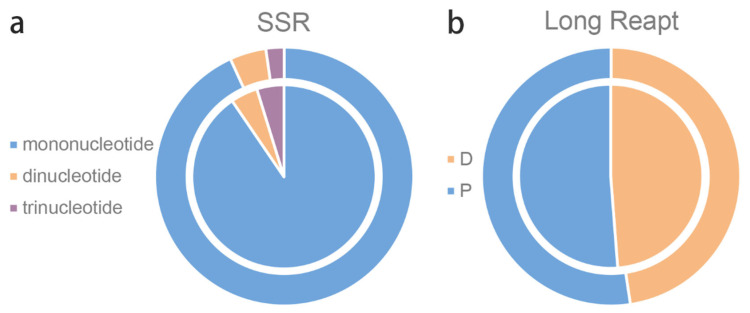
SSRs and long repeats in *O. taihangensis* and *O. longilobus.* (**a**) Type of SSRs detected in two species. (**b**) Long repeats classification of two species. The outer circle is *O. taihangensis*, and the inner circle is *O. longilobus*.

**Figure 3 genes-13-02410-f003:**
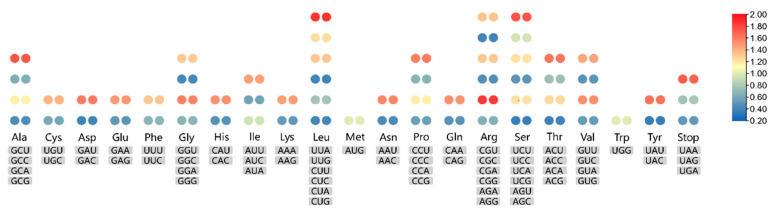
The RSCU values of all protein-coding genes for *O. taihangensis* and *O. longilobus* cp genomes. Color key: the red values indicate higher RSCU values, and the blue values indicate lower RSCU values. The left side represents *O. taihangensis*, and the right side represents *O. longilobus*.

**Figure 4 genes-13-02410-f004:**
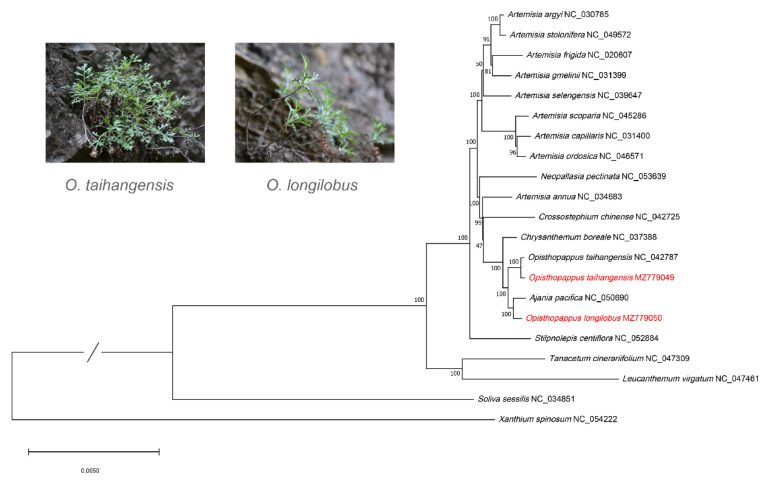
Maximum likelihood (ML) phylogenetic tree based on chloroplast genomes of subtribe Chrysantheminae. The top left corner shows a comparison of the habitat maps of the two species.

**Figure 5 genes-13-02410-f005:**
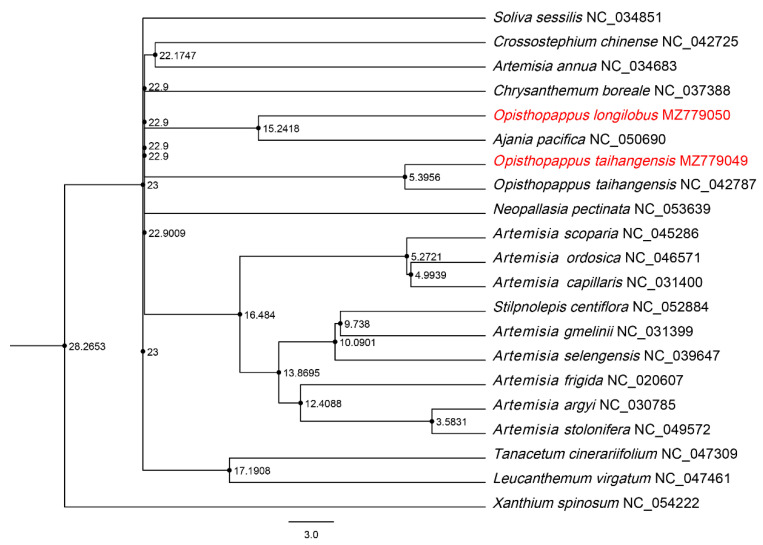
Divergence time estimation based on subtribe Chrysantheminae.

**Figure 6 genes-13-02410-f006:**
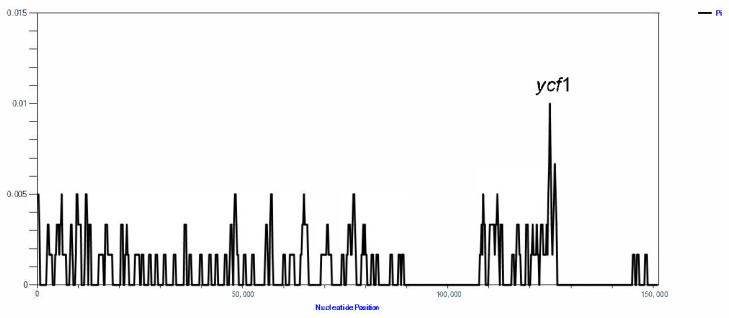
Sliding window analysis of the entire chloroplast genome of *Opisthopappus* species (window length: 1000 bp; step size: 300 bp). X-axis: position of the window; Y-axis: nucleotide diversity of each window.

**Figure 7 genes-13-02410-f007:**
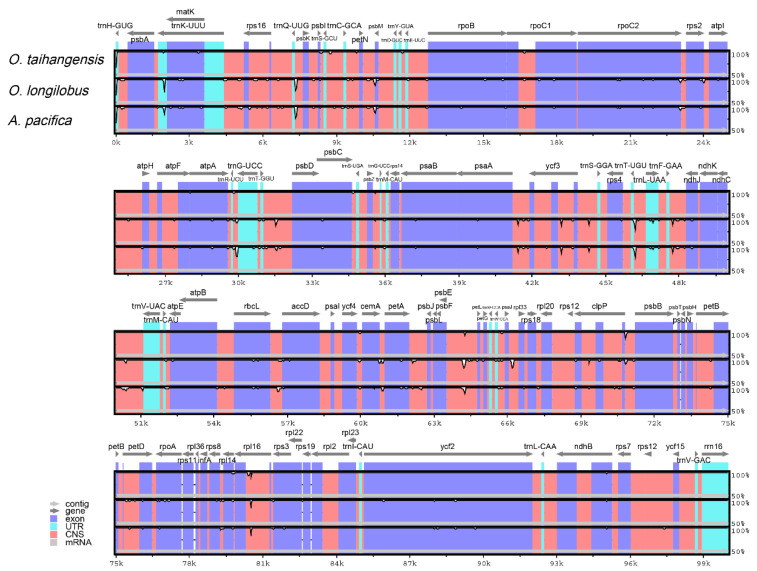
Comparison of four chloroplast genomes using *O. taihangensis* (NC_042787) annotation as a reference. The vertical scale indicates the percentage of identity, ranging from 50 to 100%. The horizontal axis indicates the coordinates within the chloroplast genome. Genome regions are color-coded as exons, introns, and intergenic spacer (IGS).

**Figure 8 genes-13-02410-f008:**
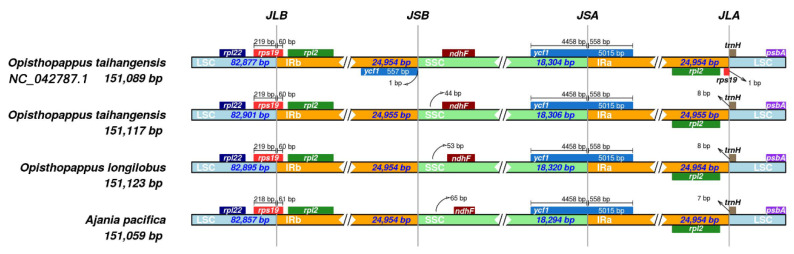
Comparative analysis of chloroplast genomic boundaries of four species cp genome.

**Table 1 genes-13-02410-t001:** Summary of the plastome features for *O. taihangensis* and *O. longilobus*. LSC: large single-copy region; SSC: small single-copy region; IR: inverted repeat.

-	Features	*O. taihangensis*	*O. longilobus*
Accession Number	-	MZ779049	MZ779050
Length/bp	Total	151,117	151,123
	LSC	82,901	82,895
	SSC	18,306	18,320
	IR	24,955	24,954
GC content/%	Total	37.46	37.44
	LSC	35.54	35.52
	SSC	30.82	30.8
	IR	43.07	43.08
No. of gene	Total	133	133
	Protein coding	88	88
	tRNA	37	37
	rRNA	8	8

**Table 2 genes-13-02410-t002:** Gene contents of *O. taihangensis* and *O. longilobus* cp genome.

Gene Function	Group Of Genes	Gene Names	Amount
rRNA	rRNA genes	*rrn*16S (×2), *rrn*23S (×2), *rrn*5S (×2), *rrn*4.5S (×2)	8
tRNA	tRNA genes	*trn*A-UGC (×2), *trn*C-GCA, *trn*D-GUC, *trn*E-UUC (×3), *trn*F-GAA, *trn*M-CAU (×4), *trn*G-GCC, *trn*H-GUG, *trn*K-UUU, *trn*L-CAA (×2), *trn*L-UAA, *trn*L-UAG, *trn*M-CAU, *trn*N-GUU (×2), *trn*P-UGG, *trn*Q-UUG, *trn*R-ACG (×2), *trn*R-UCU, *trn*S-CGA, *trn*S-GCU, *trn*S-GGA (×2), *trn*T-GGU, *trn*T-UGU, *trn*V-GAC(×2), *trn*W-CCA, *trn*Y-GUA	37
Self-replication	Small subunit of ribosome	*rps*11, *rps*12 (×2), *rps*14, *rps*15 (×2), *rps*16, *rps*18, *rps*19, *rps*2, *rps*3, *rps4*, *rps*7 (×2), *rps*8	15
	Large subunit of ribosome	*rpl*14, *rpl*16, *rpl*2 (×2), *rpl*20, *rpl*22, *rpl*23 (×2), *rpl*32, *rpl*33, *rpl*36	11
	DNA dependent RNA polymerase	*rpo*A, *rpo*B, *rpo*C1, *rpo*C2	4
Photosynthesis	Subunits of NADH-dehydrogenase	*ndh*A, *ndh*B(×2), *ndh*C, *ndh*D, *ndh*E, *ndh*F, *ndh*G, *ndh*H, *ndh*I, *ndh*J, *ndh*K	12
	Subunits of photosystem Ⅰ	*psa*A, *psa*B, *psa*C, *psa*I, *psa*J	5
	Subunits of photosystem Ⅱ	*psb*A, *psb*B, *psb*C, *psb*D, *psb*E, *psb*F, *psb*I, *psb*J, *psb*K, *psb*L, *psb*M, *psb*N, *psb*T, *psb*Z, *ycf*3	15
	Subunits of cytochrome b/f complex	*pet*A, *pet*B, *pet*D, *pet*G, *pet*L, *pet*N	6
	Subunits of ATP synthase	*atp*A, *atp*B, *atp*E, *atp*F, *atp*H, *atp*I	6
	Large subunit of rubisco	*rbc*L	1
Other genes	Maturase	*mat*K	1
	Protease	*clp*P	1
	Envelope membrane protein	*cem*A	1
	Subunit of Acetyl-CoA-carboxylase	*acc*D	1
	c-type cytochrome synthesis gene	*ccs*A	1
	Translational initiation factor	*inf*A	1
Unknown	Conserved open reading frames	*ycf*1, *ycf*15(×2), *ycf*2 (×2), *ycf*4	6
Total			133

**Table 3 genes-13-02410-t003:** Comparison of the cp genome features between NC_042787 and MZ779049.

Features		NC_042787	MZ779049
Length/bp		151089	151117
No. of gene	Total	132	133
	Protein coding	87	88
	tRNA	37	37
	rRNA	8	8

## Data Availability

All sequences used in this study are in the form of attachments. We have submitted this part of the data to NCBI but have not yet released it. At present, we have provided it to the journal and reviewers as an attachment and urge NCBI to release it as soon as possible. The dataset generated and or analyzed during the current study is deposited in Genbank with accession numbers: MZ779049 and MZ779050.

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
