# Peer review of "Complete Chloroplast Genome Sequence of Endangered Species in the Genus Opisthopappus C. Shih: Characterization, Species Identification, and Phylogenetic Relationships"

_genes, 2022, doi:10.3390/genes13122410_

Round 1

Reviewer 1 Report

Zhang et al. sequenced, assembled, and annotated the complete chloroplast genomes of two species in the Opisthopappus genus (Opisthopappus taihangensis and Opisthopappus longilobus) and compared them with other chloroplast genomes. The study is fine, and the results might be helpful for evolutionary studies. However, I would suggest some revisions as follows:

1- The complete chloroplast genome of Opisthopappus taihangensis was previously submitted with accession number "NC_042787" and published by Gu et al. (2019). It is already there! I am wondering why the authors did not use this genome. Can you explain this? Also, the authors should add a section to conduct a comprehensive comparison between NC_042787 and MZ779049.

2- The figure resolution of the circular gene map of the complete chloroplast genomes for two species in figure 1 is poor. Please fix this.

3- The text in section 2.2 is in bold. Please fix this.

Author Response

Dear Reviewer,

We would like to express our sincere thanks to you for giving us an opportunity to revise our manuscript entitled “Complete chloroplast genome sequence of endangered species in the genus Opisthopappus C. Shih: characterization, species identification, and phylogenetic relationships” (ID: genes-2067450). We have revised according to the reviewers’ constructive comments, which are marked in green in the paper. Below, please find our point-by-point responses to the reviewers’ comments.

We have read the referee’s comments very carefully and we have revised, updated, and added the content, sentences, words, and references of the article according to the comments.

Again, we would like to express our great appreciation to reviewers for comments on our paper and we have revised our manuscript.

Reviewer 2 Report

The manuscript describe  a threatened plant, and your efforts to reveal genetic information about cp sequences. I noticed many phrases need to revised and linguistically corrected. Besides, many abbreviations are mentioned directly without previous full word were written. I highlighted the confused phrases and phrases that I asked to be clarified, corrected and / or explained.

For references, I noticed two style format you use. I expect to be all unified.

Regards

Author Response

(The authors gave the same response as above.)
